# Peripheral Neutrophil-to-Lymphocyte Ratio in Bronchiectasis: A Marker of Disease Severity

**DOI:** 10.3390/biom12101399

**Published:** 2022-09-30

**Authors:** Miguel Ángel Martinez-García, Casilda Olveira, Rosa Girón, Marta García-Clemente, Luis Máiz-Carro, Oriol Sibila, Rafael Golpe, Raúl Méndez, Juan Luis Rodríguez Hermosa, Esther Barreiro, Concepción Prados, Juan Rodríguez López, David de la Rosa

**Affiliations:** 1Servicio de Neumología e Instituto de Investigación La Fe (IISLAFE), Hospital Universitario y Politécnico La Fe, 46026 Valencia, Spain; 2Centro de Investigación Biomédica en Red de Enfermedades Respiratorias, Instituto de Salud Carlos III, 28220 Madrid, Spain; 3Servicio de Neumología, Instituto de Investigación Biomédica de Málaga (IBIMA), Hospital Regional Universitario de Málaga, Universidad de Málaga, 29010 Málaga, Spain; 4Servicio de Neumología, Instituto de Investigación Sanitaria, Hospital Universitario de la Princesa, 28006 Madrid, Spain; 5Servicio de Neumología, Instituto de investigación biosanitaria del Principado de Asturias (ISPA), 33011 Oviedo, Spain; 6Servicio de Neumología, Hospital Ramón y Cajal, 28034 Madrid, Spain; 7Servicio de Neumología, Hospital Clínic, IDIBAPS, 08036 Barcelona, Spain; 8Servicio de Neumología, Hospital Lucus Augusti, 27003 Lugo, Spain; 9Pulmonary Department, Research Institute of Hospital Clínico San Carlos (IdISSC), Faculty of Medicine, University Complutense of Madrid, 28040 Madrid, Spain; 10Servicio de Neumología, Hospital del Mar-IMIM, UPF, CIBERES, 08003 Barcelona, Spain; 11Servicio de Neumología, Hospital La Paz, 28046 Madrid, Spain; 12Servicio de Neumología, Hospital San Agustín, Avilés, 33410 Asturias, Spain; 13Servicio de Neumología, Hospital Santa Creu i Sant Pau, 08041 Barcelona, Spain

**Keywords:** neutrophil, lymphocyte, bronchiectasis, severity, exacerbations

## Abstract

Most patients with bronchiectasis have a predominantly neutrophilic inflammatory profile, although other cells such as lymphocytes (as controllers of bronchial inflammation) and eosinophils also play a significant pathophysiological role. Easy-to-interpret blood biomarkers with a discriminative capacity for severity or prognosis are needed. The objective of this study was to assess whether the peripheral neutrophil-to-lymphocyte ratio (NLR) is associated with different outcomes of severity in bronchiectasis. A total of 1369 patients with bronchiectasis from the Spanish Registry of Bronchiectasis were included. To compare groups, the sample was divided into increasing quartiles of NLR ratio. Correlations between quantitative variables were established using Pearson’s P test. A simple linear regression (with the value of exacerbations as a quantitative variable) was used to determine the independent relationship between the number and severity of exacerbations and the NLR ratio. The area under the curve (AUC)-ROC was used to determine the predictive capacity of the NLR for severe bronchiectasis, according to the different multidimensional scores. Mean age: 69 (15) years (66.3% of women). The mean NLR was 2.92 (2.03). A higher NLR was associated with more severe bronchiectasis (with an especially significant discriminative power for severe forms) according to the commonly used scores (FACED, E-FACED and BSI), as well as with poorer quality of life (SGRQ), more comorbidities (Charlson index), infection by pathogenic microorganisms, and greater application of treatment. Furthermore, the NLR correlated better with severity scores than other parameters of systemic inflammation. Finally, it was an independent predictor of the incident number and severity of exacerbations. In conclusion, the NLR is an inexpensive and easy-to-measure marker of systemic inflammation for determining severity and predicting exacerbations (especially the most severe) in patients with bronchiectasis.

## 1. Introduction

In both phenotypic and endotypic terms, bronchiectasis is a complex and heterogeneous disease [1,2,3,4,5] defined by a pathophysiological substrate of chronic airway inflammation in which neutrophils play a fundamental role [6,7]. However, lymphocytes [8], eosinophils [9,10,11], and other inflammatory cells [12] have also been seen to play an important role in some circumstances. This inflammation, along with various products of microorganisms (mainly bacterial proteolytic substances), is what ultimately causes the irreversible damage to the bronchial wall and airway dilation that characterize bronchiectasis and explain its symptoms [7]. Furthermore, a variable percentage of patients, especially those with greater severity, present a certain degree of systemic inflammation of a very heterogeneous nature [3,13,14,15,16,17], although some authors have found homogeneous groups based on parameters of cell or molecular counts in peripheral blood [18,19,20].

Given that the neutrophilic and proteolytic products generated by bronchiectasis (particularly neutrophil elastase [NE]) seem to play a fundamental role in the disease’s genesis and progression, as has been shown in some studies [21,22], efforts have been made to find anti-inflammatory treatments to block this NE and thereby prevent the damage caused by the inflammation itself [23,24]. Moreover, some strains of lymphocytes also play a crucial role in the immune system’s control of the inflammation found in bronchiectasis [6,7].

It seems important, therefore, to find new, easily accessible and interpretable biomarkers that are associated with various important outcomes in bronchiectasis and that may mark more specific targets for future treatments. In this respect, the blood neutrophil-to-lymphocyte ratio (NLR) (two cells that, as already mentioned, form a fundamental part of bronchial inflammation in bronchiectasis patients) has been analyzed in diseases of the airway, e.g., COPD [25,26,27,28], and the lungs, e.g., pneumonia [29], cystic fibrosis [30], and interstitial disease [31] as well as some types of cancer [32,33] and cardiovascular disease [34]. The NLR has been associated with various prognostic outcomes and degrees of severity, even in patients with COVID-19 infection [35], but even though this marker is cheap and easy to obtain, its value in bronchiectasis patients [36,37] has been little studied. Our working hypothesis is that the NLR will also be associated, as in the case of COPD (another disease largely characterized by predominantly neutrophilic bronchial inflammation), with various parameters of severity and prognosis in bronchiectasis. Therefore, the objective of this study was to analyze the associations between the NLR and various parameters of severity and prognosis in bronchiectasis, as well as other outcomes of interest, in a large series of steady-state patients.

## 2. Methods

### 2.1. Study Design

This was a prospective, longitudinal, observational, multicentre study of a cohort derived from the Spanish Bronchiectasis Registry (RIBRON) [38]. The registry was started in February 2015 and involves 43 centres from all over Spain. It includes adult patients (at least 18 years old) diagnosed with bronchiectasis by means of high-resolution computerized tomography, with related clinical symptoms.

### 2.2. Patients

At the time of the analysis, data were available on 2039 patients included from February 2015 to December 2019. 

Inclusion criteria were: analytical data available at entry, including systemic inflammatory markers (both molecules and cells, especially total number of neutrophils and lymphocytes) in conditions of clinical stability (defined as at least 4 weeks free of an exacerbation period) and complete data on exacerbations during the first year of follow-up. The main criterion for exclusion was a diagnosis of cystic fibrosis (CF). 

All the patients gave their written informed consent at their participating centre, as provided by the local ethics committee affiliated with the registry (number: 001-2012. Hospital Josep Trueta, Girona). 

### 2.3. Variables and Definitions

The following variables were used for the purposes of this study: baseline general and anthropometric data; aetiology; severity scores (FACED [39], E-FACED [40] and Bronchiectasis Severity Index [BSI] [41]; comorbidities (Charlson index); lung function; treatments; quality of life (St George Respiratory Questionnaire [SGRQ]; clinical, analytical, radiological and microbiological data; and the number and severity of incident exacerbations during the first year of follow-up. The FACED score is formed by 5 variables and divides bronchiectasis into mild (0–2 points), moderate (3–4 points) or severe (5–7 points); the E-FACED score is formed by 6 variables and divides bronchiectasis into mild (0–3 points), moderate (4–6 points) or severe (7–9 points), and the BSI is formed by 9 variables and divides bronchiectasis into mild (0–4 points), moderate (5–8 points) or severe (at least 9 points). Finally, the Charlson index is used to quantify comorbidities. Nineteen conditions were included in the index. Each condition was assigned a weight from 1 to 6, based on the estimated 1-year mortality.

An exacerbation was defined (when the registry was created) as a worsening of the typical symptoms of bronchiectasis: cough, dyspnoea, haemoptysis, increase in the volume or purulence of the sputum, chest pain and sibilance with an evolution of more than 24 h for which antibiotic treatment was required. An exacerbation was considered mild-moderate when the patient needed oral antibiotics, and severe in cases of hospital admission or when intravenous antibiotic treatment was required in either a hospital or home setting [42]. Exacerbator patients were defined as those with at least three exacerbations per year or two mild-to-moderate exacerbations plus at least one hospitalization [43,44]. Chronic bronchial infection (CBI) was defined as the presence of three or more consecutive cultures positive for the same potentially pathogenic microorganisms (PPM) [42,45]. 

Finally, the neutrophil-to-lymphocyte ratio (NLR) was calculated as the quotient between the absolute neutrophil count and the absolute lymphocyte count, in both cases as cells/µL of peripheral blood.

### 2.4. Statistical Analysis

Data were tabulated using the mean ± standard deviation or median (interquartile range) for quantitative data that, respectively, followed or did not follow a normal distribution. The normality of the distribution was analysed using the Kolmogorov-Smirnov test. The qualitative data were tabulated according to the percentage with respect to the total value. Since the NLR did not follow a normal distribution, the sample was divided into increasing quartiles for comparison between groups, using ANOVA or the corresponding nonparametric tests if necessary. Correlations between quantitative variables were established using Pearson’s *p* test or Spearman’s test, depending on the variable distribution. A simple linear regression (with the number of exacerbations as a quantitative variable) was used to determine the independent relationship between the number and severity of exacerbations and the NLR ratio. A logistic regression was used to determine the risk of entering the group of exacerbator patients, adjusted for clinically significant variables according to the authors’ criteria. All the analyses were also adjusted for clinically significant variables, including bronchiectasis severity, age, gender and treatments, according to the authors’ criteria.

The area under the curve (AUC)-ROC was used to determine the predictive capacity of the NLR for severe bronchiectasis, according to the different multidimensional scores. The C statistic was used to make comparisons between ROC curves. The statistical package SPSS Inc. 20 was used.

## 3. Results

Of the 2039 patients included in the RIBRON at the time of the analysis, 127 were excluded due to CF, and a further 543 patients were excluded due to a lack of the baseline total numbers of neutrophils and lymphocytes or a lack of data on the number of exacerbations during the first year of follow-up. Thus, 1369 patients were eventually included. A comparison of the baseline characteristics of bronchiectasis patients not included in the study (n = 543) with those who were included did not reveal any significant differences (data not shown).

The mean age was 69 ± 15 years (66.3% women), and 37.2% of the bronchiectasis presented a post-infectious aetiology. The Charlson index was 1.8 (1.5), and the total SGRQ was 30.9 ± 17.5, while 11.8% of the patients presented concomitant COPD, and 57% presented *Pseudomonas aeruginosa* (PA) isolations (22.3% as CBI). The mean predicted FEV1 was 75 ± 24.8%. The FACED, E-FACED and BSI mean values were 2.1 ± 1.7, 2.7 ± 2.2 and 7.5 ± 4.9, respectively. The mean number of exacerbations during the first year of follow-up was 1.52 ± 1.6, and that of hospitalizations was 1.05 ± 1.3, while 31.6% of the patients were defined as frequent exacerbators. The mean NLR was 2.92 ± 3.03 (range 0.11–25.8), with a median of 2 (IQR: 1.43–3.02).

Table 1 shows the comparison of the four groups formed according to the increasing quartiles of NLR. It can be seen that the higher the NLR, the more significant the increase in age, comorbidities (Charlson index), dyspnoea, CBI (in particular by PA), the number and severity of exacerbations in the year after admission in the registry and the use of bronchodilator and anti-inflammatory treatments (inhaled corticosteroids and macrolides). Furthermore, the higher the NLR, the greater the decrease in the quality of life (only in the SGQR symptom domain), the FEV1, the percentage of women and 0_2_ saturation. In short, increased severity, as measured by the three multidimensional scores (BSI, FACED and E-FACED), corresponded with higher NLRs. No significant differences were observed in terms of radiological extension, domains of activity, impact of the SGRQ or the aetiology of bronchiectasis (although there was a higher but nonsignificant percentage of patients with COPD and a lower percentage of idiopathic bronchiectasis in the quartile with the highest NLR).

### 3.1. The NLR and Bronchiectasis Severity Indexes

The NLR not only correlated with the scores used, it was also capable of distinguishing between the different severities (mild, moderate and severe) into which these scores are divided. The correlations between the NLR figures and the FACED, E-FACED and BSI (r = 0.32; r = 0.36 and r = 0.33, respectively) were higher than those found for blood neutrophil counts (r = 0.24; r = 0.29; r = 0.29, respectively) and blood lymphocyte counts (r = −0.06; r = −0.04, r = −0.02, respectively), as well as for other acute phase reactants such as C-reactive protein (CRP), fibrinogen and platelets, without being significant (Figure 1). 

The prognostic capacity to identify patients with the highest severity score (BSI-Figure 2A, E-FACED-Figure 2B and FACED-Figure 2C) in the three severity scales was significantly higher for the NLR than for the separate absolute numbers of neutrophils and lymphocytes. When comparing the AUC-ROCs for the three scores between those provided by the NLR and those provided by the absolute number of neutrophils, they were statistically higher for the latter in all three scores (BSI: 0.64 vs 0.71; *p* = 0.001; E-FACED: 0.63 vs 0.71, *p* = 0.001, FACED: 0.64 vs 0.68, *p* = 0.016).

### 3.2. Correlations between NLR and Other Bronchiectasis Variables

Table 2 shows the correlation between NLR and some clinical, functional, radiological, microbiological and biological variables. NLR significantly correlated with almost all the variables that marked a higher degree of severity in bronchiectasis. The highest correlations were seen with dyspnoea (r = 0.23, *p* < 0.0001), the number (r = 0.21, *p* < 0.001) and severity (r = 0.22, *p* < 0.001) of exacerbations, the degree of airflow obstruction (r = −0.25; *p* < 0.001), PA isolation (r = 0.27; *p* < 0.001) and fibrinogen levels (r = 0.28; *p* < 0.001).

### 3.3. Prognostic Value of Exacerbations

The quantitative NLR was related to both the number and severity of incident exacerbations occurring during the first year after admission. These values were more significant in the relationship between the NLR and severe exacerbations (Table 3). In the same way, the higher the NLR (according to quartiles), the greater the risk of belonging to the group of exacerbator patients, but only the comparison between the highest quartile and the first quartile (control group) was statistically significant, i.e., for NLR greater than 3.03 (Table 4). All calculations were adjusted for the severity of bronchiectasis (according to the three scores), gender, presence of COPD and corticosteroid or macrolide treatment (those variables that were already part of the multidimensional scores, such as age, were not included as covariates to avoid duplication).

## 4. Discussions

According to our results, the NLR was closely related to bronchiectasis severity (especially the severest forms) measured by the three validated scores, and it correlated better with multidimensional scores than with other blood inflammatory biomarkers such as CRP, fibrinogen or the number of platelets. Moreover, the NLR was associated with poor clinical, functional and quality-of-life outcomes, as well as with a higher probability of PA infection. This marker demonstrated a good prognostic value for incident exacerbations (especially severe ones) and for the risk of entering the group of exacerbator patients during the first year of follow-up.

Inflammation and immunity response play a critical role in the pathophysiology of bronchiectasis [6,7,46].

On the one hand, both the number and function of neutrophils are crucial in the control of infection by PPM, and on the other hand, lymphocytes represent adaptive immunity against not only bacterial infection but also the hyperexpression of neutrophilic inflammation, which can further damage the bronchial wall on account of the proteolitic products secreted by neutrophils. In fact, decreases in circulating lymphocytes coincide with recent infection. Therefore, neutrophils and lymphocytes can also be jointly regulated through complex mechanisms [47,48,49,50].

In recent years, some authors have found that the blood NLR could be a good, cheap and easy surrogate marker of bronchial inflammation [51]. It has been observed that a high NLR is associated, in the general population, with higher mortality [52,53]. It has been also related to some cancers [32,33] and cardiovascular [34], systemic and respiratory disorders [30,31]. In COPD, a high NLFR has been associated with a greater number of exacerbations, dyspnoea, poor lung function, higher severity scores on radiological scales and the BODE Index, various cardiovascular factors, bacterial colonization and even higher mortality [25,26,27,28].

To date, only two small retrospective studies in children have analysed the value of the NLR in bronchiectasis patients. Nacaroglu et al. [36] observed an increase in the NLR in 50 children with bronchiectasis compared with a control group, especially during exacerbation processes. In contrast, Coban et al. [54] found, in a group of 117 patients with steady-state bronchiectasis, no relationship between the NLR and the severity of the disease, although they did observe a correlation between this marker and other markers of systemic inflammation, such as CRP. However, Georgakopoulou et al. [37] observed, in 80 exacerbated patients with bronchiectasis, a transient increase in the NLR that particularly correlated with the isolation of PPM in sputum, albeit without any linear correlation with CRP values.

Given that some authors suggest that the amount of certain inflammatory cells such as neutrophils in peripheral blood can be a good surrogate for the inflammation that occurs at the pulmonary level [48], the NLR could be hypothesized as having more value than the separate percentages or absolute numbers of neutrophils or lymphocytes for assessing the severity or prognosis of inflammatory airway diseases, including bronchiectasis.

According to our results, the NLR is an easy-to-measure and cheap biomarker with an excellent capacity to identify those patients with more severe bronchiectasis [39,40,41], more clinical activity [53] and a greater number of exacerbations [55,56,57,58], all of which are important prognostic variables in bronchiectasis. Furthermore, the NLR proved, as in the case of COPD [25], to be related to poor clinical, functional and quality-of-life parameters and the presence of PA infection.

In terms of aetiology, those patients with COPD presented higher NLRs than those with idiopathic and post-infectious bronchiectasis. This finding was possibly related to the higher degree of inflammation and severity, as well as the greater frequency of a prognosis of the overlap syndrome with bronchiectasis already observed in other studies [59]. In fact, the NLR has also been shown to be related to the severity of COPD and has demonstrated its prognostic value for exacerbations in these patients [25,26,27,28]. However, the ability to discriminate the severity of bronchiectasis and the prognosis of the number and severity of NLR exacerbations was not dependent on the presence of COPD, since the results did not vary significantly when COPD patients were excluded from the analysis.

Lastly, the correlation between the NLR and severity scores was higher than that found with other markers of systemic inflammation such as fibrinogen, CRP and platelets, which suggests that this marker could provide additional information that is easy to obtain and monitor in those more severe patients who will suffer from a greater number and severity of exacerbations.

Among the strengths of our study, the large number of patients included is particularly outstanding. Among its limitations is the fact that no prognostic results of NLR are presented in a follow-up of more than one year. Moreover, we have not presented data on mortality as an outcome, and there are no biopsy data available to confirm that the blood NLR faithfully reflects the effects of airway inflammation, in contrast to some other studies [60]. Finally, we decided to divide our sample into quartiles, in line with other similar studies (reference), since the NTL variable did not follow a normal distribution. We have not established an optimal cut-off point for NLR since we analysed various outcomes (exacerbations and three different severity scores) and each of these would probably have its own separate optimal cut-off point.

In conclusion, the NLR correlates with greater functional and clinical severity of bronchiectasis, presenting a good prognostic capacity to discern those patients with a greater number and severity of future exacerbations. Future studies are necessary to confirm our results via international validation as well as to assess whether the addition of this parameter to the already validated multidimensional scores would provide new information and whether this peripheral marker is associated with other important long-term outcomes or mortality in patients with bronchiectasis.

## Figures and Tables

**Figure 1 biomolecules-12-01399-f001:**
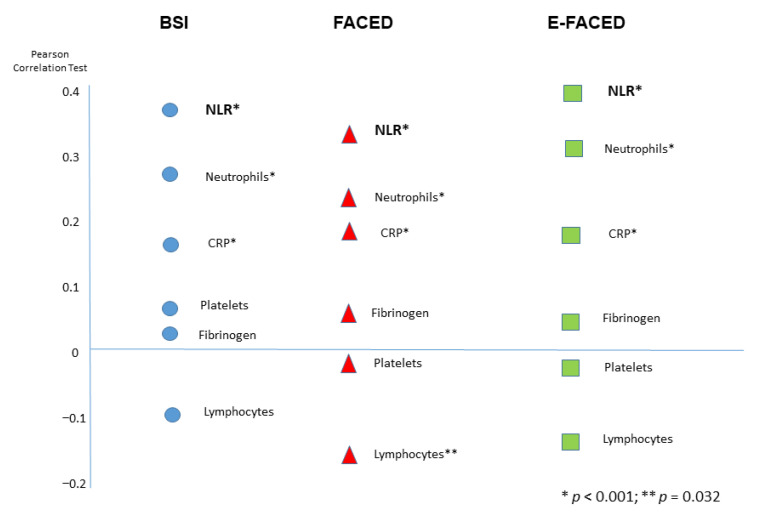
Correlations between different systemic inflammation markers (including the neutrophil-to-lymphocyte ratio) and severity scales in bronchiectasis. BSI: Bronchiectasis Severity Index, NLR: neutrophil-to-lymphocyte ratio; CRP. C-reactive protein.

**Figure 2 biomolecules-12-01399-f002:**
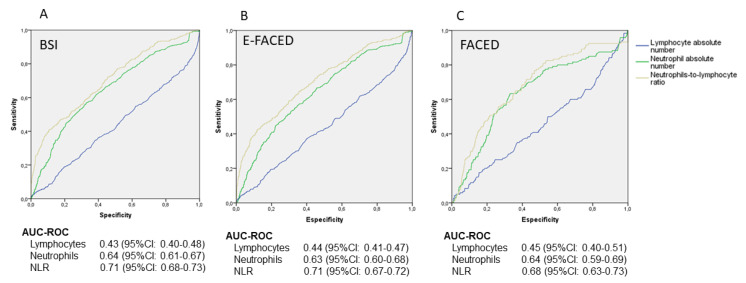
Prognostic capacity of the neutrophil-to-lymphocyte ratio for severe bronchiectasis, according to the Bronchiectasis Severity Index (**A**), E-FACED (**B**) and FACED (**C**). BSI: Bronchiectasis Severity Index, NLR: neutrophil-to-lymphocyte ratio; AUC-ROC: area under curve-ROC.

**Table 1 biomolecules-12-01399-t001:** Differentiating characteristics of the quartiles of the neutrophil-to-lymphocyte ratio.

Variable	Q1 (0.11–1.43)	Q2 (1.44–2)	Q3 (2.01–3.02)	Q4 (3.03–25.8)	*p* ANOVA
Number	342	349	336	342	--
NLR	1.1 ± 0.24	1.7 ± 0.17	2.5 ± 0.29	6.5 ± 4.4	<0.0001
Age, yrs	68 ±15	67.1 ±15	69.5 ± 14.4	71.4 ±15.8	0.001
Gender (% women)	74%	66%	65%	60%	0.002
BMI, Kg/m^2^	25.4 ± 4.6	26.2 ± 5	26 ± 4.9	25.9 ± 5.2	0.223
Smoking habit (packs.year)	28.7 ±26	30.9 ± 25.1	33.2 ± 28.1	34.7 ± 26.4	0.232
Charlson Index	1.6 ± 1.1	1.7 ± 1.49	1.9 ± 1.6	2.1 ± 1.7	0.0001
Etiology % COPD Post-infectious Idiopathic	10% 37% 19%	11% 35% 21%	11% 38% 15%	16% 39% 14%	0.107 0.646 0.061
Type Cylindrical Varicose Cystic	90 18 18	92 20 19	86 29 23	80 27 27	0.015 0.041 0.033
Dyspnoea (mMRC)	1.6 ± 0.8	1.7 ± 0.9	1.8 ± 0.9	2.1 ± 1	0.0001
Purulent sputum, %	48%	47%	48%	53%	0.506
Daily sputum production, %	22%	22%	24%	27%	0.359
FEV1, %	81.1 ± 23	79 ± 23.2	64.6 ± 23.9	65.7 ± 26.3	0.0001
Pulmonary lobes	2.7 ±1.5	2.7 ±1.4	2.9 ±1.4	2.9 ±1.4	0.334
O2 saturation, %	96.2 ± 2.1	95.7 ± 2.7	95.8 ± 3.1	95.1 ± 3.3	0.0001
Eosinophils, %	3.6 ±3.3	3.2 ±3.6	3.1 ± 2.8	1.7 ± 1.8	0.0001
A1AT levels (pg/mL)	137 ± 40	132 ± 39	132 ± 34	139 ± 42	0.296
SGRQ Symptoms Activity Impact Total	31.1 ± 25 28.9 ± 19.8 15.7 ± 15.9 23.1 ± 17.4	37 ± 23 37.3 ± 25.8 17.1 ± 18.8 27.5 ± 19.4	37.7 ± 12.4 41.7 ± 18.8 20.9 ± 15.5 30 ± 13.7	50.6 ± 24 45.4 ± 25.1 25.3 ± 15.2 35.6 ± 17.2	0.041 0.177 0.250 0.130
PPM isolations, % *S. aureus* *H. influenzae* *P. aeruginosa*	97% 21% 27% 41%	97% 16% 20% 54%	96% 8% 24% 59%	98% 7% 17% 70%	0.001 0.021 0.0001
PPM by CBI, % CBI by *P. aeruginosa* CBI (other PPM)	11% 20%	19% 28%	27% 35%	32% 37%	0.0001 0.0001
Exacerbations	0.9 ± 1.2	1.1 ± 1.6	1.2 ± 1.3	1.3 ± 1.7	0.041
Hospitalizations	0.4 ± 1.3	0.5 ± 1	0.6 ± 1.4	1.1 ± 1.7	0.0001
Previous pneumonia episode, %	42%	35%	44%	49%	0.003
FACED score Mild Moderate Severe	1.6 ± 1.4 72.8% 26.6% 0.6%	1.7 ± 1.5 71.6% 27.2% 1.1%	2.2 ± 1.7 63.1% 31.8% 5.1%	2.8 ± 1.9 46.8% 41.5% 11.7%	<0.0001 0.0001 0.0001 0.0001
E-FACED score Mild Moderate Severe	2.1 ± 1.8 81.9% 16.4% 1.8%	2.2 ± 1.9 77.4% 19.8% 2.9%	2.7 ± 2.1 71.4% 23.5% 5.1%	3.8 ± 2.5 47.1% 37.4% 15.5%	<0.0001 0.0001 0.0001 0.0001
BSI Mild Moderate Severe	6.2 ± 4.1 43.5% 34.2% 22.2%	6.7 ± 4.4 39.3% 30.2% 30.5%	7.6 ± 4.7 30.9% 37.1% 32.1%	9.8 ± 5.6 19.8% 26.6% 53.6%	0.0001 0.0001 0.0001 0.0001
Bronchodilators, %	65%	69%	74%	76%	0.004
Inhaled steroids, %	42%	47%	51%	58%	0.001
Macrolides, %	7%	8%	7%	10%	0.655

BSI: Bronchiectasis Severity Index; CI. confidence interval; NLR: neutrophil-to-lymphocyte ratio; OR: odds ratio; Q: quartile; BMI: body mass index; COPD: chronic obstructive pulmonary disease; SGRQ: Saint George Respiratory Questionnaire; PPM: potentially pathogenic microorganisms; *S. aureus*: *Staphilococcus aureus*; *H. influenzae*: *Haemophilus influenzae*; *P. aeruginosa: Pseudomonas aeruginosa*; CBI: chronic bronchial infection; mMRC: modified Medical Research Council; A1AT: alpha-1 antitrypsin levels.

**Table 2 biomolecules-12-01399-t002:** Correlation between NLR value and general, clinical, functional, biological, microbiological, radiological and therapeutic parameters.

Variable	Correlation Coefficient; *p* Value
General and clinical variables Age, yrs Gender BMI, Kg/m^2^ Dyspnoea, (mMRC) Charlson index Number of exacerbations Number of hospitalizations	0.09; *p* = 0.001 −0.10; *p* = 0.001 0.03; *p* = 0.34 0.23; *p* < 0.001 0.13; *p* = 0.001 0.21; *p* < 0.001 0.22; *p* < 0.001
Functional variables 02 sat, % FEV1/FVC, % FVC, % FEV1, %	−0.13; *p* = 0.001 −0.21; *p* = 0.001 −0.15; *p* = 0.001 −0.25; *p* < 0.001
Peripheral biomarkers Neutrophils, % Lymphocytes, % Eosinophils, % Fibrinogen CRP Platelets	0.56; *p* < 0.001 −0.58; *p* < 0.001 −0.22; *p* = 0.001 0.28; *p* < 0.001 0.12; *p* = 0.001 0.07; *p* = 0.016
Microbiological variables PA isolation HI isolation SA isolation	0.27; *p* < 0.001 0.08; *p* = 0.08 0.17; *p* = 0.001
Radiological parameters Number of lobes Cystic bronchiectasis	0.07; *p* = 0.009 0.10; *p* = 0.004
Treatment Macrolides Inhaled corticosteroids Bronchodilators	0.02; *p* = 0.451 0.11, *p* = 0.001 0.03; *p* = 0.081

BMI: body mass index; mMRC: Modified Medical Research Council; CRP: C-reactive protein; PA: Pseudomonas aeruginosa; HI: Haemophilus influenza; SA: Staphylococcus aureus.

**Table 3 biomolecules-12-01399-t003:** Simple linear regression. Prognostic value of the neutrophil-to-lymphocyte ratio for exacerbations and hospitalizations according to three different models adjusted for gender and treatments, based on the various severity scores for bronchiectasis.

	β Coeff	*p*	95%CI
Model 1: BSI			
Number of exacerbations Number of hospitalizations	0.05 0.04	0.012 0.0001	0.01–0.08 0.02–0.06
Model 2: E-FACED			
Number of exacerbations Number of hospitalizations	0.04 0.12	0.017 0.0001	0.01–0.09 0.08–0.16
Model 3: FACED			
Number of exacerbations Number of hospitalizations	0.02 0.09	0.041 0.001	0.01–0.05 0.03–0.11

BSI: Bronchiectasis Severity Index; CI. Confidence Interval; β coeff: beta coefficient.

**Table 4 biomolecules-12-01399-t004:** Logistical regression to evaluate the risk of belonging to the group of exacerbator patients, according to the quartile, in the neutrophil-to-lymphocyte ratios in three models adjusted for gender and treatments, based on the severity scores for bronchiectasis.

	NLR Quartiles	*p*	OR	95%CI
Model 1: BSI	Q1 Q2 Q3 **Q4**	*Control* 0.25 0.13 **0.003**	0.8 1.3 **1.7**	0.6–1.1 0.7–1.4 **1.2–2.3**
Model 2: E-FACED	Q1 Q2 Q3 **Q4**	*Control* 0.23 0.17 **0.001**	0.7 1.2 **1.8**	0.7–1.2 0.8–1.5 **1.3–2.2**
Model 3: FACED	Q1 Q2 Q3 **Q4**	*Control* 0.14 0.11 **0.021**	0.7 1.1 1.6	0.8–1.1 0.6–1.4 **1.1–1.9**

BSI: Bronchiectasis Severity Index; CI. confidence interval; NLR: neutrophil-to-lymphocyte ratio; OR: odds ratio; Q: quartile.

## Data Availability

The data presented in this study are available on request from the corresponding author.

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
