# Peer review of "Peripheral Neutrophil-to-Lymphocyte Ratio in Bronchiectasis: A Marker of Disease Severity"

_biomolecules, 2022, doi:10.3390/biom12101399_

Round 1

Reviewer 1 Report

The manuscript entitled "Peripheral Neutrophil-to-lymphocyte Ratio in Bronchiectasis: a Marker of Disease Severity" done by Miguel Angel Martinez-Garcia et al. is very interesting in the field of bronchiectasis.This study aimes to assess whether the peripheral neutrophil-to-lymphocyte ratio (NLR) is associated with different outcomes of severity in bronchiectasis. The results of study are based on 1,369 patients with bronchiectasis from the Spanish Registry of Bronchiectasis. The authors show that a higher value of NLR was associated with greater severity of bronchiectasis by using the commonly used scores (FACED, E-FACED and BSI); and it is an independent predictor of the incident number and severity of exacerbations. They conclude that the NLR is an inexpensive and easy-to-measure marker of systemic inflammation for determining severity and predicting exacerbations (especially the most severe) in patients with bronchiectasis. However, there are some minor comments to be clarified for improving the manuscript before publication: 1.Methods: - The scoring methods for FACED, E-FACED, BSI should be described in the same section - The Charson index should be described  - The criteria of bronchiectasis should be defined 2. Results: - The specific correlation between NLR with other clinical, functional and biologic features of bronchiectasis should be done (Figures). - Continuous parameters should be presented by mean +/- SD

Author Response

Reviewer 1.

POINT-BY-POINT RESPONSES

Biomolecules (ISSN 2218-273X)

biomolecules-1876655

Article

Peripheral Neutrophil-to-lymphocyte Ratio in Bronchiectasis: a Marker of Disease Severity.

Miguel Angel Martinez-Garcia * , Casilda Olveira , Rosa Mª Girón , Marta García Clemente , Luis Maiz-Carro , Oriol Sibila , Rafael Golpe , Raúl Méndez , Juan Luis Rodríguez Hermosa , Esther Barreiro , Concepción Prados , Juan Rodríguez , David De la Rosa

Cellular Biochemistry

The manuscript entitled "Peripheral Neutrophil-to-lymphocyte Ratio in Bronchiectasis: a Marker of Disease Severity" done by Miguel Angel Martinez-Garcia et al. is very interesting in the field of bronchiectasis.This study aimes to assess whether the peripheral neutrophil-to-lymphocyte ratio (NLR) is associated with different outcomes of severity in bronchiectasis. The results of study are based on 1,369 patients with bronchiectasis from the Spanish Registry of Bronchiectasis. The authors show that a higher value of NLR was associated with greater severity of bronchiectasis by using the commonly used scores (FACED, E-FACED and BSI); and it is an independent predictor of the incident number and severity of exacerbations. They conclude that the NLR is an inexpensive and easy-to-measure marker of systemic inflammation for determining severity and predicting exacerbations (especially the most severe) in patients with bronchiectasis.

However, there are some minor comments to be clarified for improving the manuscript before publication:

Q1.Methods: - The scoring methods for FACED, E-FACED, BSI should be described in the same section - The Charson index should be described  - The criteria of bronchiectasis should be defined

R1. We have now described the scoring methods from FACED, E-FACED, BSI and Charlson index, as well as the bronchiectasis diagnosis criteria in the methods section of the paper.

Q2. Results: - The specific correlation between NLR with other clinical, functional and biologic features of bronchiectasis should be done (Figures). - Continuous parameters should be presented by mean +/- SD

R2. Thank for your comment. We have added this information in the results section in a new table. We have also used mean +/-SD as required.

Reviewer 2 Report

The authors investigated the association between the blood neutrophil-to-lymphocyte ratio (NLR) and various parameters of severity and prognosis in patients with bronchiectasis. These analyzes were performed on the large number (a total of 1369 patients) of patients with bronchiectasis from the Spanish Bronchiectasis Registry. The authors concluded the NLR was closely related to bronchiectasis severity (especially the severest forms) measured by the three validated scores, and correlated better with multidimensional scores than other blood inflammatory biomarkers. Furthermore, the NLR correlates with a greater functional and clinical severity of bronchiectasis, presenting a good prognostic capacity to discern those patients with a greater number and severity of future exacerbations. 

In view of the lack of studies on the NLR of patients with bronchiectasis, the present study is considered a very interesting and well-written manuscript. However, this study has a few limitations that do not show the survival data and validation study. Therefore, this manuscript needs some supplement points to improve the quality of the article and the understanding of international readers. 

1. Further discussion of these limitations which mentioned above is needed 

2. The authors analyzed according to the classification into four groups by the increasing quartiles of NLR values.  Why didn’t you analyze the results using the optimal cutoff values of NLR? These optimal cutoff values for the NLR can be determined by ROC curve, maximally selected rank statistics or other methods. The authors would like to discuss this point.  

3. Please provide overall survival (survival curve) of the total study population according to the four NLR groups.

4. In the abstract, it is necessary to briefly mention the study methods.

5. In the legend of figure 1, 2, “*hay que cambiar "Lymphocites" a "Lymphocytes" en la figura, and *Hay que cambiar  "Neutrophils-to" a "Neutrophil-to" en la figura.” What do these mean? 

6. As already mentioned in the present paper, the inflammation is what ultimately causes the irreversible damage to the bronchial wall and airway dilation that characterize bronchiectasis and explain its symptoms. In addition, the cumulative previous studies indicate that blood NLR is a cost-effective prognostic factor in patients with various diseases. Therefore, the analysis of bronchoalveolar lavage in patients with bronchiectasis can be very important. Thus, please refer to the recent following article in the article

Ryu WK, et al.. A Preliminary Study on the Prognostic Impact of Neutrophil to Lymphocyte Ratio of the Bronchoalveolar Lavage Fluid in Patients with Lung Cancer. Diagnostics (Basel) 2021;11 

Author Response

Reviewer 2

POINT-BY-POINT RESPONSES

Biomolecules (ISSN 2218-273X)

biomolecules-1876655

Article

Peripheral Neutrophil-to-lymphocyte Ratio in Bronchiectasis: a Marker of Disease Severity.

Miguel Angel Martinez-Garcia * , Casilda Olveira , Rosa Mª Girón , Marta García Clemente , Luis Maiz-Carro , Oriol Sibila , Rafael Golpe , Raúl Méndez , Juan Luis Rodríguez Hermosa , Esther Barreiro , Concepción Prados , Juan Rodríguez , David De la Rosa

Cellular Biochemistry

The authors investigated the association between the blood neutrophil-to-lymphocyte ratio (NLR) and various parameters of severity and prognosis in patients with bronchiectasis. These analyzes were performed on the large number (a total of 1369 patients) of patients with bronchiectasis from the Spanish Bronchiectasis Registry. The authors concluded the NLR was closely related to bronchiectasis severity (especially the severest forms) measured by the three validated scores, and correlated better with multidimensional scores than other blood inflammatory biomarkers. Furthermore, the NLR correlates with a greater functional and clinical severity of bronchiectasis, presenting a good prognostic capacity to discern those patients with a greater number and severity of future exacerbations. 

In view of the lack of studies on the NLR of patients with bronchiectasis, the present study is considered a very interesting and well-written manuscript. However, this study has a few limitations that do not show the survival data and validation study. Therefore, this manuscript needs some supplement points to improve the quality of the article and the understanding of international readers. 

Q1. Further discussion of these limitations which mentioned above is needed 

R1. We have added these limitations as required.

Q2. The authors analyzed according to the classification into four groups by the increasing quartiles of NLR values.  Why didn’t you analyze the results using the optimal cutoff values of NLR? These optimal cutoff values for the NLR can be determined by ROC curve, maximally selected rank statistics or other methods. The authors would like to discuss this point.  

R2. We have used four groups (quartiles) since the NLR variable did not follow a normal distribution, and to optimize the number of subjects in each group under consideration we included 1,369 patients, i.e., over 350 per group. It is true that we could have used 3 or 5 groups, but we finally decided to use 4 groups, in keeping with other papers related to NLR:  (COVID-19: Higaki et al. Sci Rep 2022; Depression: Meng F et al, J Affect Disord 2022; Cardiovascular disease: Mannarino et l. Biofactors 2022; Lung cancer: Grieshober L, Cancer Causes Control 2021; General mortality: Song M, et al. Sci Rep 2021; ARDS: Li W, et al. Shock 2029; COPD: Lee et al. PLoS One 2016, etc…).

Finally, please take into account that we analyzed various outcomes (exacerbations and three different severity scores) and each of these would probably have its own separate optimal cutoff point. We have addressed this point, as required, in the discussion section.

Q3. Please provide overall survival (survival curve) of the total study population according to the four NLR groups.

R3. Thanks for the suggestion. Unfortunately, we can´t add these data to this paper since we have already written another paper on the impact of NLR on mortality and sent it to another journal. We hope the reviewer understands the situation.

Q4. In the abstract, it is necessary to briefly mention the study methods.

R4. Done.

Q5. In the legend of figure 1, 2, “*hay que cambiar "Lymphocites" a "Lymphocytes" en la figura, and *Hay que cambiar  "Neutrophils-to" a "Neutrophil-to" en la figura.” What do these mean? 

R5. Thanks for the comment and we apologize for the mistakes. We have now amended them.

Q6. As already mentioned in the present paper, the inflammation is what ultimately causes the irreversible damage to the bronchial wall and airway dilation that characterize bronchiectasis and explain its symptoms. In addition, the cumulative previous studies indicate that blood NLR is a cost-effective prognostic factor in patients with various diseases. Therefore, the analysis of bronchoalveolar lavage in patients with bronchiectasis can be very important. Thus, please refer to the recent following article in the article

Ryu WK, et al.. A Preliminary Study on the Prognostic Impact of Neutrophil to Lymphocyte Ratio of the Bronchoalveolar Lavage Fluid in Patients with Lung Cancer. Diagnostics (Basel) 2021;11 

R6. Thanks for the comment. We have added some comments on this topic.

Reviewer 3 Report

In this current study, Martínez-García et al analyzed clinical data from 1969 patients from SEPAR (Spanish Society of Pneumology) Spanish Bronchiectasis Registry (RIBRON), emphasizing the association between peripheral neutrophil-to-lymphocyte ratio (NLR) and severity in bronchiectasis outcomes. Authors focused on three different scores of bronchiectasis severity (FACED, E-FACED and Bronchiectasis Severity Index), and compared these values to the corresponding NLR values.

The significance of the study observations greatly diminished by the fact that similar analyses have already been reported earlier. In fact, the NLR ratio is implicated as a biomarker for various pathological conditions.

Authors are requested to address the following concerns.

Suggestions

1. Authors must justify the selection of the four quartiles to divide the patients. The four quartiles represent different NLR ranges: 0.11 to 1.43, 1.44 to 2, 2.01 to 3.02, and finally 3.03 to 25.8. Please explain the rationale for these ranges. Also, authors are requested to report the number of patients categorized under each of these quartiles.

2. Generally, three main types of bronchiectasis are recognized, namely cylindrical, varicose and cystic. Is it possible to identify these three groups and perform a comparison among them in terms of NLR ratio?

3. The four quartiles considered here differ significantly in terms of age and the percentage of female patients. Authors should consider normalizing the values and examine whether the changes still persist.

4. The correlation values mentioned in the Results section “The NLR and bronchiectasis severity indexes” do not match with the Figure 1’s values. Please edit accordingly.

5. It is not clear what the authors are reporting in the Results section “The NLR and aetiology”. From the methods section, it seemed that authors removed several patients from the analysis. But here they have compared COPD patients and indicated that when 162 COPD patients were removed from the analysis, there was no difference.

Other comments

1. Results, line 129-131, “A comparison of the… differences (data not shown)”, please edit this statement.

2. Various quantitation is presented as numerical value with another measurement within the parenthesis. Please indicate what these values mean.

3. Some comments in Spanish language are there in the manuscript.

4. The manuscript requires editing for language use.

Author Response

Reviewer 3

POINT-BY-POINT RESPONSES

Biomolecules (ISSN 2218-273X)

biomolecules-1876655

Article

Peripheral Neutrophil-to-lymphocyte Ratio in Bronchiectasis: a Marker of Disease Severity.

Miguel Angel Martinez-Garcia * , Casilda Olveira , Rosa Mª Girón , Marta García Clemente , Luis Maiz-Carro , Oriol Sibila , Rafael Golpe , Raúl Méndez , Juan Luis Rodríguez Hermosa , Esther Barreiro , Concepción Prados , Juan Rodríguez , David De la Rosa

Cellular Biochemistry

In this current study, Martínez-García et al analyzed clinical data from 1969 patients from SEPAR (Spanish Society of Pneumology) Spanish Bronchiectasis Registry (RIBRON), emphasizing the association between peripheral neutrophil-to-lymphocyte ratio (NLR) and severity in bronchiectasis outcomes. Authors focused on three different scores of bronchiectasis severity (FACED, E-FACED and Bronchiectasis Severity Index), and compared these values to the corresponding NLR values.

The significance of the study observations greatly diminished by the fact that similar analyses have already been reported earlier. In fact, the NLR ratio is implicated as a biomarker for various pathological conditions.

Thanks for the comment. Although it is true that the NTL ratio is implicated as a biomarker for various pathological conditions, there is very little literature related to NCF bronchiectasis, as we explained in the paper, and our study is by far the largest on this topic.

Authors are requested to address the following concerns.

Suggestions

Q1. Authors must justify the selection of the four quartiles to divide the patients. The four quartiles represent different NLR ranges: 0.11 to 1.43, 1.44 to 2, 2.01 to 3.02, and finally 3.03 to 25.8. Please explain the rationale for these ranges. Also, authors are requested to report the number of patients categorized under each of these quartiles.

R1. Thanks for the comment. We have used four groups (quartiles) since the NLR variable did not follow a normal distribution, and to optimize the number of subjects in each group under consideration we included 1,369 patients, i.e., over 350 per group.  It is true that we could have used 3 or 5 groups, but we finally decided to use 4 groups, in keeping with other papers related to NLR: (COVID-19: Higaki et al. Sci Rep 2022; Depression: Meng F et al, J Affect Disord 2022; Cardiovascular disease: Mannarino et l. Biofactors 2022; Lung cancer: Grieshober L, Cancer Causes Control 2021; General mortality: Song M, et al. Sci Rep 2021; ARDS: Li W, et al. Shock 2029; COPD: Lee et al. PLoS One 2016, etc…).

Finally, we have added the number of subjects in each quartile in the table 1

Q2. Generally, three main types of bronchiectasis are recognized, namely cylindrical, varicose and cystic. Is it possible to identify these three groups and perform a comparison among them in terms of NLR ratio?

R2. Yes, thanks for the comment. We can do this and we have added the comparison to Table 1.

Q3. The four quartiles considered here differ significantly in terms of age and the percentage of female patients. Authors should consider normalizing the values and examine whether the changes still persist.

R3. As stated on page 8, “All calculations were adjusted for the severity of bronchiectasis (according to the values of the three scores), gender, presence of COPD and corticosteroid or macrolide treatment (those variables that were already part of the multidimensional score, such as age, were not included as covariates to avoid duplication)". We have now also pointed this out in the statistical analysis section.

Q4. The correlation values mentioned in the Results section “The NLR and bronchiectasis severity indexes” do not match with the Figure 1’s values. Please edit accordingly.

R4. Thanks for the comment and sorry for our mistake. It has now been amended.

Q5. It is not clear what the authors are reporting in the Results section “The NLR and aetiology”. From the methods section, it seemed that authors removed several patients from the analysis. But here they have compared COPD patients and indicated that when 162 COPD patients were removed from the analysis, there was no difference.

R5. The reviewer is right and we also think that this point is confusing. So, I have decided to delete it since there were no statistically differences between aetiologies (post-infectious, idiopathic and COPD) with respect to the NLR values in Table 1.  

Other comments

Q6. Results, line 129-131, “A comparison of the… differences (data not shown)”, please edit this statement.

R6. Thanks for the comment. The amendment has been made.

Q7. Various quantitation is presented as numerical value with another measurement within the parenthesis. Please indicate what these values mean.

R7. Thanks for the comment. We have not used mean +/- standard deviation to avoid confusion

Q8. Some comments in Spanish language are there in the manuscript.

R8. Yes, it was a mistake. Thanks for the comment.

Q9. The manuscript requires editing for language use.

R9. We have checked the English accordingly.

Round 2

Reviewer 2 Report

The authors adequately addressed the requested comments in the revised manuscript, except R3. Thank you for your effort.